# Curcumin-Loaded Solid Lipid Nanoparticles Bypass P-Glycoprotein Mediated Doxorubicin Resistance in Triple Negative Breast Cancer Cells

**DOI:** 10.3390/pharmaceutics12020096

**Published:** 2020-01-24

**Authors:** Gamal-Eldein Fathy Abd-Ellatef, Elena Gazzano, Daniela Chirio, Ahmed Ragab Hamed, Dimas Carolina Belisario, Carlo Zuddas, Elena Peira, Barbara Rolando, Joanna Kopecka, Mohamed Assem Said Marie, Simona Sapino, Sohair Ramadan Fahmy, Marina Gallarate, Abdel-Hamid Zaki Abdel-Hamid, Chiara Riganti

**Affiliations:** 1Department of Oncology, University of Torino, via Santena 5/bis, 10126 Torino, Italy; gamalology@yahoo.com (G.-E.F.A.-E.); elena.gazzano@unito.it (E.G.); dimascarolina.belisario@unito.it (D.C.B.); carlo.zuddas@unito.it (C.Z.); joanna.kopecka@unito.it (J.K.); 2Therapeutic Chemistry Department, Pharmaceutical and Drug Industries Research Division, National Research Centre, 33 El Bohouth St., Dokki, Giza 12622, Egypt; marina.gallarate@unito.it (M.G.); abdelhamidzaki@hotmail.com (A.-H.Z.A.-H.); 3Department of Drug Science and Technology, University of Torino, via P. Giuria 9, 10125 Torino, Italy; daniela.chirio@unito.it (D.C.); elena.peira@unito.it (E.P.); barbara.rolando@unito.it (B.R.); simona.sapino@unito.it (S.S.); 4Chemistry of Medicinal Plants Department, Biology Unit, Central Laboratory of Pharmaceutical and Drug Industries Research Division, National Research Centre, 33 El Bohouth St., Dokki, Giza 12622, Egypt; n1ragab2004@yahoo.com; 5Zoology Department, Faculty of Science, Cairo University, Gamaa Street, Dokki, Giza 12622, Egypt; massemmarie@yahoo.com (M.A.S.M.); sohairfahmy@gmail.com (S.R.F.)

**Keywords:** solid lipid nanoparticles, curcumin, P-glycoprotein, doxorubicin resistance, triple negative breast cancer

## Abstract

Multidrug resistance (MDR) is a critical hindrance to the success of cancer chemotherapy. The main thing responsible for MDR phenotypes are plasma-membranes associated with adenosine triphosphate (ATP) Binding Cassette (ABC) drug efflux transporters, such as the P-glycoprotein (Pgp) transporter that has the broadest spectrum of substrates. Curcumin (CURC) is a Pgp inhibitor, but it is poorly soluble and bioavailable. To overcome these limitations, we validated the efficacy and safety of CURC, loaded in biocompatible solid lipid nanoparticles (SLNs), with or without chitosan coating, with the goal of increasing the stability, homogeneous water dispersibility, and cellular uptake. Both CURC-loaded SLNs were 5–10-fold more effective than free CURC in increasing the intracellular retention and toxicity of doxorubicin in Pgp-expressing triple negative breast cancer (TNBC). The effect was due to the decrease of intracellular reactive oxygen species, consequent inhibition of the Akt/IKKα-β/NF-kB axis, and reduced transcriptional activation of the Pgp promoter by p65/p50 NF-kB. CURC-loaded SLNs also effectively rescued the sensitivity to doxorubicin against drug-resistant TNBC tumors, without signs of systemic toxicity. These results suggest that the combination therapy, based on CURC-loaded SLNs and doxorubicin, is an effective and safe approach to overcome the Pgp-mediated chemoresistance in TNBC.

## 1. Introduction

Multidrug resistance (MDR), i.e., a cross-resistance to a broad variety of anticancer drugs unrelated to structure and activity, produces chemotherapy failure and tumor progression [1]. One of the most studied mechanisms of MDR is the overexpression of drug efflux pumps belonging to the adenosine triphosphate (ATP) Binding Cassette (ABC) transporters family, characterized by a broad and overlapping spectrum of substrates [2,3]. The main ABC transporters clinically associated with the development of MDR are P-glycoprotein (Pgp/ABCB1), MDR Related Proteins (MRPs/ABCCs), and Breast Cancer Resistance Protein (BCRP/ABCG2) [4,5].

Triple negative breast cancer (TNBC) is an aggressive and invasive subtype of breast cancer that accounts for 10–20% of all breast cancers [6]. Chemotherapy based on anthracyclines, such as doxorubicin, and taxanes are the main treatment options for patients with TNBC. After neo-adjuvant chemotherapy, i.e., pre-surgery chemotherapy aimed at reducing tumor mass, sees about 30% to 40% patients achieve a complete response with no histological evidence of disease at the time of surgery [7]. However, the onset of drug resistance due to the presence of Pgp reduces the efficacy of neoadjuvant and adjuvant chemotherapy in many cases [8,9]. Improving the success rate of anthracycline-based chemotherapy in TNBC is still an unmet need.

The latest frontier in Pgp inhibition is the co-administration of antineoplastic agents with pumps inhibitors [10,11]. The co-encapsulation within liposomes or biocompatible nanoparticles has been experimented to improve pharmacokinetic and pharmacodynamic features of Pgp inhibitors [12].

Curcumin (CURC) is a secondary metabolite isolated from the turmeric of *Curcuma longa,* which has several biological activities, including inhibiting Pgp activity and expression [13,14]. Since CURC is a highly lipophilic drug, different CURC-loaded nano-formulations were developed in order to enhance its solubility, stability, specificity, tolerability, cellular uptake/internalization, efficacy, and therapeutic index [15,16]. Solid lipid nanoparticles (SLNs) are interesting nanocarriers to be exploited in CURC delivery. Biocompatibility, particle size (below 400 nm), chemical and mechanical stability, easy functionalization, and enhanced delivery of bioactive lipophilic molecules represent the most advantageous properties of SLN [17,18,19]. The solid lipid matrix protects entrapped lipophilic drugs from chemical degradation and enhances their physical stability. SLN ameliorates the half-life of drugs in the systemic circulation, modulates their release kinetics, and increases the therapeutic efficacy of drugs used in anticancer therapy [20,21]. Moreover, SLN surfaces can be decorated by several agents. Among the coating materials, chitosan (CS) is a non-toxic, biocompatible, and biodegradable polymer, and has been proven to control the release of drugs. Its fair solubility in aqueous media avoids the use of organic solvents during SLN preparation, and once added to the synthesized SLNs, it does not require further SLN purification [22]. Owing to its hydrophilic character, amphiphilic CS derivatives are commonly used with a lipid matrix like that of SLNs [23].

The aim of this work is to test and mechanistically investigate the properties of two different formulations of CURC-loaded SLNs (with and without chitosan) as examples of biocompatible nanomaterials that are able to improve CURC delivery to TNBC cells, enhance its property of inhibiting Pgp, and reverse doxorubicin resistance.

## 2. Materials and Methods

### 2.1. Chemicals and Materials

Trilaurin (TL), ethyl acetate (EA), benzyl alcohol (BenzOH), butyl lactate (BL), sodium taurocholate (NaTC), Pluronic^®^ F68, 1,2 propanediol, CURC, cholesterol, chitosan and Sepharose^®^ CL 4B, doxorubicin were purchased from Sigma Chemicals Co. (St. Louis, MO, USA). Epikuron^®^200 (lecithin-phosphatydilcoline 92%) was from Cargill (Minneapolis, MN, USA), Cremophor^®^RH 60 (PEG-60 hydrogenated castor oil) from BASF (Ludwigshafen, Germany). The plastic ware for cell cultures was obtained from Falcon (Becton Dickinson, Franklin Lakes, NJ, USA). The electrophoresis reagents were from Bio-Rad Laboratories (Hercules, CA, USA). The protein content of cell lysates was assessed with the BCA kit from Sigma Chemicals Co. Deionized water was obtained by a MilliQ system (Millipore, St. Louis, MO, USA). Unless specified otherwise, all reagents were purchased from Sigma Chemicals Co.

### 2.2. SLN Preparation and Characterization

SLNs were prepared by the “cold dilution of microemulsion” method. This technique involved the preparation of an oil/water (O/W) microemulsion (µE) with a disperse oil phase consisting of a solution of a solid lipid dissolved in a partially water miscible solvent. The solvent and the external phase, consisting of water, were mutually saturated at 25 ± 2 °C for 2 h in order to ensure the initial thermodynamic equilibrium of both liquids, before using them in µE formulation. Following water dilution of the µE, the organic solvent was removed from the disperse phase and was extracted by dissolution into the continuous phase, with the following SLN precipitations. In a previous work, a preliminary screening was performed on several compositions obtained with biocompatible GRAS (Generally Recognized As Safe) ingredients [23]. On the basis of this screening, we used EA or BL (as partially water-soluble organic solvents), TL or cholesterol/stearoyl chitosan (CS), prepared as reported in References [23,24], as lipid matrixes. Briefly, two different solutions of (a) TL in EA were used; and (b) cholesterol/CS in BL were used as lipid phases, while Epikuron^®^200, NaTC and Cremophor^®^RH60 were employed as surfactant/co-surfactant. EA/BL and water (W) were mutually pre-saturated (s-EA, s-BL, s-W) before use. The resulting µEs were then diluted by 2% *w/w* Pluronic^®^F68 aqueous solution to precipitate SLNs. Two different CURC-loaded SLNs were prepared, with or without CS: CURC-CS-SLN and CURC-SLN, respectively, as reported in References [23,24].

In CURC-CS-SLN preparation, CURC was solubilized with cholesterol in s-BL; the continuous aqueous phase, surfactant and co-surfactant were then added to form a clear system. The final step was the precipitation of CURC-CS-SLN obtained by µE water dilution and their purification by gel chromatography, which was necessary to separate CURC-loaded particles from the free drug. CURC-SLN was prepared in a similar way—by dissolving CURC in s-EA. The main difference between the two formulations concerned the SLN surface characteristics: The surface of CURC-SLN was lipophilic, while the surface of CURC-CS-SLN was hydrophilic, because the lipophilic chains of CS fit inside the SLN lipid matrix, while -OH groups were exposed outward, conferring hydrophilic characters to the nanoparticle surface.

The composition of CURC-SLN and CURC-CS-SLN is reported in Table 1.

Dimensional analysis, drug entrapment and overtime stability were reported in Reference [24]. In particular, both SLNs types had mean diameters lower than 200 nm. CURC entrapment efficiency was in the 70–75% range of the total CURC amount.

### 2.3. Cell Lines 

Human MCF-7 cells, human TNBC MDA-MB-231 and murine mammary cancer JC cells (syngeneic with balb/C mice) were purchased from ATCC^®^ (Manassas, VA, USA). Cells were maintained in RPMI-1640 media supplemented with 10% *v/v* fetal bovine serum (FBS), 1% *v/v* penicillin-streptomycin, 1% *v/v* L-glutamine.

### 2.4. Co-Culture Models

In co-culture assays, 1 × 10^5^ human monocytic THP-1 cells (ATCC^®^), seeded in RPMI-1640 medium containing 10% FBS, were stimulated with 500 nM phorbol-12 myristate 13-acetate (PMA) for 24 h in order to favor their differentiation into macrophages [25]. Floating THP-1 cells were removed by washing the dishes twice with PBS; attached cells—i.e., macrophage-differentiated cells—were rinsed with fresh medium. 1 × 10^6^ MDA-MB-231 cells (i.e., a 1:10 ratio between macrophages and tumor cells) [26], were added. Co-cultures were incubated for 24 h, as indicated in the Results Section. Next, cells were detached, macrophages were removed using the Pan Monocyte Isolation Kit (Miltenyi, Bergisch Gladbach, Germany), as per the manufacturer’s instructions. All the assays were performed on MDA-MB-231 cells separated after the co-culture.

### 2.5. Doxorubicin Accumulation

5 × 10^5^ cells were incubated, as reported in the Results Section, washed twice with PBS, and gently scraped and centrifuged at 13,000× *g* for 5 min at 4 °C. The amount of intracellular doxorubicin was detected fuorimetrically using a HT Synergy 96-well micro-plate reader (Bio-Tek Instruments, Winooski, VT, USA), using excitation and emission wavelengths of 475 and 553 nm, respectively [27]. Fluorescence was converted in nmoles doxorubicin/mg cell proteins using a calibration curve with serial dilutions of doxorubicin in 1:1 *v/v* ethanol/HCl 0.3 N (i.e., the same buffer used to extract doxorubicin from cells). We preliminarily verified that, in these conditions, the fluorescence spectrum of doxorubicin was the same as that obtained from cell lysates.

### 2.6. Cytotoxicity

The release of lactate dehydrogenase (LDH) in the extracellular medium, considered an index of doxorubicin cytotoxicity [27], was measured spectrophotometrically [28]. 50 µL of the culture medium were centrifuged at 12,000× *g* for 15 min, and diluted in 0.2 mL of 82.3 mM triethanolammine phosphate hydrochloride (TRAP, pH 7.6). LDH activity was measured in 200 µL of medium, by adding 5 mM NADH and 20 mM pyruvic acid, measuring the change in absorbance at 340 nm with a HT Synergy 96-well micro-plate reader, for 6 min. The reaction kinetics was linear. The results were expressed as µmoles NAD^+^/min/mg cell proteins.

### 2.7. Rhodamine 123 Efflux

Rhodamine 123 accumulation, which is inversely related to its efflux, was used as a second index of Pgp activity [29]. The intracellular rhodamine 123 content was detected fluorimetrically, using a HT Synergy 96-well micro-plate reader. The results were expressed as nmoles/mg cell proteins.

### 2.8. Immunoblotting

Cells were rinsed with ice-cold lysis buffer (50 mM Tris, 10 mM EDTA, 1% *v/v* Triton-X100), supplemented with the protease inhibitor cocktail set III (80 μM aprotinin, 5 mM bestatin, 1.5 mM leupeptin, and 1 mM pepstatin; Calbiochem, San Diego, CA, USA), 2 mM phenylmethylsulfonyl fluoride and 1 mM Na_3_VO_4_, then sonicated and centrifuged at 13,000× *g* for 10 min at 4 °C. 20 μg protein extracts were subjected to SDS-PAGE and probed with antibodies for: Anti-Pgp/ABCB1 (Calbiochem), anti-MRP1/ABCC1 (Abcam, Cambridge, UK), anti-BCRP/ABCG2 (Santa Cruz Biotechnology Inc., Santa Cruz, CA, USA), anti-(phosphoSer473)Akt (Cell Signalling Technology, Danvers, MA, USA), anti-Akt (Cell Signalling Technology), anti-(phosphoSer176/180)IKKα/β (Cell Signalling Technology), anti-IKKα/β (Cell Signalling Technology), and anti-IkBα (Santa Cruz Biotechnology Inc.), followed by a peroxidase-conjugated secondary antibody (Bio-Rad Laboratories). The membranes were washed with Tris-buffered saline-Tween 0.1% *v/v* solution, and the proteins were detected by enhanced chemiluminescence (Bio-Rad Laboratories). To check the equal control loading in lysates, samples were probed with an anti-β-tubulin (Santa Cruz Biotechnology Inc.) antibody.

### 2.9. Flow Cytometry

Cells were harvested, washed twice in PBS, detached with cell dissociation solution (Sigma Chemical Co.) and re-suspended in a culture medium containing 5% *v/v* FBS. Samples were washed with 0.25% *w/v* PBS-bovine serum albumin (BSA), and incubated with the primary antibody for an anti-ABCB1/Pgp antibody (clone C219, Abcam) for 45 min at 4 °C. After washing with PBS-BSA 1% *w/v* twice, cells were incubated with a secondary fluorescein isothiocyanate (FITC)-conjugated antibody (Sigma Chemical Co.) for 30 min at 4 °C. After washing twice with PBS-BSA 1% *w/v* and fixing in paraformaldehyde 2% *w/v* for 5 min at room temperature, samples were analyzed by a Guava^®^ easyCyte flow cytometer (Millipore), using the InCyte software (Millipore). Control experiments included incubation of cells with a non-immune isotypic antibody, followed by the secondary antibody.

### 2.10. Quantitative Real Time-PCR (qRT-PCR)

Total RNA was extracted by phenol/chloroform method. 1 μg RNA was reverse-transcribed using the iScript Reverse Transcription Supermix kit (Bio-Rad Laboratories), according to the manufacturer’s instruction. 25 ng cDNA were amplified with 10 μL IQ^TM^ SYBR Green Supermix (Bio-Rad Laboratories). Primers were designed with the qPrimer Depot software (http://primerdepot.nci.nih.gov/): *Pgp/ABCB1*: 5′-TGCTGGAGCGGTTCTACG-3′,5′-ATAGGCAATGTTCTCAGCAATG-3′; *MRP1/ABCC1*: 5′-CATTCAGCTCGTCTTGTCCTG-3′; 5′-GGATTAGGGTCGTGGATGGTT-3′; *BCRP/ABCG2*: 5′-GTTTCAGCCGTGGAAC-3′; 5′-CTGCCTTTGGCTTCAAT-3′; *S14*: 5′-CGAGGCTGATGACCTGTTCT-3′, and 5′-GCCCTCTCCCACTCTCTCTT-3′. qRT-PCR was carried out with an iQ^TM^5 cycler (Bio-Rad Laboratories). Cycling conditions were: 1 cycle of 30 s at 95 °C, 40 cycles of denaturation (15 s at 95 °C), and annealing/extension (30 s at 60 °C). The same cDNA preparation was used to quantify the genes of interest and the housekeeping gene *S*14, which was used to normalize gene expression. The relative quantitation of each sample was performed using the Gene Expression Quantitation software (Bio-Rad Laboratories). Results were expressed in arbitrary units. For each gene, the expression in untreated cells was considered to be 1.

### 2.11. Reactive Oxygen Species (ROS) Measurement

1 × 10^5^ cells were re-suspended in 0.5 mL. PBS, incubated for 30 min at 37 °C with 5 μM of the fluorescent probe 5-(and-6)-chloromethyl-2′,7′-dichlorodihydro-fluorescein diacetate-acetoxymethyl ester (DCFDA-AM), centrifuged at 13,000× *g* at 37 °C for 5 min, and re-suspended in 0.5 mL PBS, as reported previously [30]. The fluorescence of each sample, considered an index of ROS levels, was read at 492 (λ excitation) and 517 nm (λ emission), using a HT Synergy 96-well micro-plate reader. The results were expressed as nmoles/mg cell proteins.

### 2.12. Nuclear Factor-kB (NF-kB) and Hypoxia Inducible Factor-1α (HIF-1α) Activity. 

Nuclea were isolated with the Nuclear Extraction kit (Active Motif, Rixensart, Belgium), as per the manufacturer’s instructions. NF-kB activity was measured on 10 µg of nuclear proteins with the TransAM Flexi NF-kB activation kit (Active Motif), using the whole mix of antibodies provided by the kit (p50, p65, Rel-A, c-Rel, and p52), to measure to global NF-kB activity, or each antibody separately, to measure the activation of specific NF-kB components. HIF-1α activity was measured using the HIF Activation Kit (Active Motif), as per the manufacturer’s instructions. The absorbance at 450 nm was measured with a Packard EL340 microplate reader (Bio-Tek Instruments). The results were expressed as mU/mg nuclear proteins.

### 2.13. Chromatin Immunoprecipitation (ChIP)

Chromatin immunoprecipitation (ChIP) experiments were performed using the Magna ChIP A/G Chromatin Immunoprecipitation kit (Millipore), as per the manufacturer’s instructions. Samples were immunoprecipitated with 5 µg of ChIP-grade anti-p50 (Abcam) or anti-p65 (Abcam) antibodies, or with no antibody, as a blank. The immunoprecipitated DNA was then washed twice with 100 µL of elution buffer (0.1 M NaHCO_3_, 0.1% *v/v* sodium dodecyl sulfate), the crosslinking was reversed by incubating the samples at 65 °C for 6 h, then samples were incubated with proteinase K (Sigma Chemicals Co.) for 1 h at 55 °C. The DNA was eluted using the GenElute Mammalian Genomic DNA Miniprep kit (Sigma Chemicals Co.) and analyzed by qRT-PCR, as detailed above. The primer sequences of the promoter of *mdr1* gene, which encodes Pgp/ABCB1, designed with Primer3 software (http://frodo.wi.mit.edu/primer3), were: 5′-CGATCCGCCTAAGAACAAAG-3′; 5′-AGCACAAATTGAAGGAAGGAG-3′. The following primers were used to amplify the sequence of the *mdr1* promoter from 20 ng of non-immuno-precipitated genomic DNA: 5′-GACCAAGCTCTCCTTGCATC-3′ and 5′-AGGGAAGTCTGGCAGCTGTA-3′. The results were expressed as a ratio of the expression in immuno-precipitated samples and the expression in genomic samples. The relative expression of this ratio in untreated samples was considered as 1. As negative internal controls, immuno-precipitated samples were subjected to qRT-PCR with the following primers matching 10,000 bp upstream of the promoter: 5′-GTGGTGCCTGAGGAAGAGAG-3′ and 5′-GCAACAAGTAGGCACAAGCA-3′. In this condition, no qRT-PCR product was detected (data not shown).

### 2.14. In Vivo Tumor Growths and Hematochemical Parameters

1 × 10^7^ JC cells were mixed with 100 μL Matrigel and orthotopically implanted in 6 week-old female immunocompetent balb/C mice (Charles River Laboratories Italia, Calco), housed (5 per cage) under 12 h light/dark cycle, with food and drinks provided ad libitum. Tumor growth was measured daily by caliper, according to the equation (L × W^2^)/2, where L = tumor length and W = tumor width. When the tumor reached the volume of 50 mm^3^, mice (*n* = 8/group) were randomized and treated on day 1, 7, and 14 after randomization as follows: (1) vehicle group, treated with 200 µL saline solution intravenously (i.v.); (2) CURC group (cur), treated with 5 mg/kg CURC, dissolved in 200 µL water/10% *v/v* DMSO solution i.v.; (3) doxorubicin group (dox), treated with 5 mg/kg doxorubicin, dissolved in 200 µL water i.v.; (4) CURC + doxorubicin group, treated with 100 µL of water/10% *v/v* DMSO solution containing 5 mg/kg CURC + 100 µL water solution containing 5 mg/kg doxorubicin; (5) CURC-CS-SLN group, treated i.v. with 200 µL of saline solution of the indicated SLNs containing 5 mg/kg CURC; (6) CURC-CS-SLN + doxorubicin group, treated i.v. with 100 µL of saline solution of the indicated SLNs containing 5 mg/kg CURC + 100 µL water solution containing 5 mg/kg doxorubicin; (7) CURC-SLN group, treated i.v. with 100 µL of saline solution of the indicated SLNs containing 5 mg/kg CURC; (8) CURC-SLN + doxorubicin group, treated i.v with 200 µL of saline solution of the indicated SLNs containing 5 mg/kg CURC + 100 µL water solution containing 5 mg/kg doxorubicin. Tumor volumes were monitored daily. Animals were euthanized at day 21 after randomization with zolazepam (0.2 mL/kg) and xylazine (16 mg/kg) [31].

LDH, aspartate aminotransferase (AST), alanine aminotransferase (ALT), alkaline phosphatase (AP), creatinine, creatine phosphokinase (CPK) and CPK-MB, cardiac troponin I (cTnI), and T (cTnT) were measured on blood samples collected immediately after euthanasia, using commercially available kits from Beckman Coulter Inc. (Beckman Coulter, Miami, FL, USA). The animal care and experimental procedures were approved by the Bio-Ethical Committee of the Italian Ministry of Health (#122/2015-PR; 28 November 2014).

### 2.15. Statistical Analysis

All data in the text and figures are provided as means + SD. The results were analyzed by a one-way analysis of variance (ANOVA) and Tukey’s test, using GraphPad Prism (v 6.01) software and the Statistical Package for Social Science (SPSS) software (IBM SPSS Statistics v.19). *p < 0.05* was considered significant.

## 3. Results

### 3.1. CURC-Loaded SLNs Are More Effective than Free CURC in Increasing Doxorubicin Efficacy in Resistant Triple Negative Breast Cancer Cells

In this work, human TNBC MDA-MB-231 cells and murine JC cells were used. Both cell lines were characterized by high levels of Pgp compared to MCF7 cells, used as a doxorubicin-sensitive cell line [32] with undetectable levels of Pgp. MDA-MB-231 cells also had low levels of MRP1 and BCRP, which were undetectable in JC cells (Appendix A).

As shown in Figure 1a,b, free CURC increased doxorubicin retention and doxorubicin-induced cell damage at 25–50 μM. However, at these concentration, CURC was also cytotoxic without doxorubicin. In order to reduce the concentration of CURC below this toxicity threshold and to maintain its ability to increase doxorubicin accumulation, we tested CURC-SLN and CURC-CS-SLN. Indeed, we hypothesized that SLNs may allow a higher delivery of CURC within the cells, enhancing the chemosensitizing effects towards free CURC. Preliminary experiments showed that blank SLNs, i.e., SLNs without CURC, did not increase the release of LDH after 24 h in MDA-MB-231 cells if the nano-carriers were diluted at 1:100 in the culture medium (Appendix A). This dilution, which corresponded to a final concentration of 5 μM CURC, was used in all the subsequent experiments. Both CURC-SLN and CURC-CS-SLN increased doxorubicin accumulation (Figure 1c) and doxorubicin-induced the release of LDH, without being cytotoxic in the absence of doxorubicin (Figure 1d). Similar effects on doxorubicin accumulation and cytotoxicity were obtained on the highly Pgp-expressing JC cells (Figure 1e,f).

### 3.2. CURC-Loaded SLNs Decrease Pgp Activity and Expression

To clarify whether the effects of the SLNs were due to changes in Pgp expression or activity, we first measured whether the CURC released from the SLNs might have acted as inhibitors of Pgp efflux activity, evaluating the retention of rhodamine 123, another typical Pgp substrate, in MDA-MB-231 cells. In long-term assays—i.e., after a 24 h incubation of free CURC or CURC-loaded nanoformulations, followed by 20 min incubation with rhodamine 123—we observed an increased intracellular retention of rhodamine 123 in cells treated with CURC-SLN and CURC-CS-SLN (Figure 2a). These results suggest a diminished efflux of the dye via Pgp. By contrast, in short-term assays (i.e., CURC and CURC-loaded SLNs co-incubated with rhodamine 123 for 20 min), we did not detect any changes in rhodamine 123 content (Appendix A). This experimental set suggests that it is unlikely that SLNs loaded with CURC act as competitive inhibitors of Pgp.

We hypothesize that the increase in doxorubicin and rhodamine 123 retention was due to changes in Pgp expression. To investigate this issue, we first measured the amount of Pgp on the cell surface, corresponding to the active form of the protein. As shown in Figure 2b,c, CURC-SLN and CURC-CS-SLN slightly decreased the amount of surface Pgp in MDA-MB-231-Pgp cells. Doxorubicin increased Pgp, as a consequence of *mdr1* gene up-regulation in response to the drug [33]. This increase was not reversed by the free CURC or blank SLNs, but only by CURC-loaded SLNs. The changes in the surface Pgp proteins were paralleled by changes in the mRNA of Pgp. While in the absence of doxorubicin, we did not detect any significant changes in Pgp mRNA, and doxorubicin treatment very significantly increased it. CURC-SLN and CURC-CS-SLN, but not free CURC or blank SLNs, reduced such increase (Figure 2d). By contrast, neither doxorubicin nor the other treatments changed the mRNA levels of MRP1/ABCC1 (Appendix A) and BCRP/ABCG2 (Appendix A), suggesting that the effect of SLNs carrying CURC was specific for the Pgp transcription.

### 3.3. CURC-Loaded SLNs Decrease Pgp Transcription by Reducing Intracellular ROS and NF-kB Activity

We next investigated the potential mechanisms of CURC-loaded SLNs in reducing Pgp transcription after doxorubicin exposure. Doxorubicin is known to increase ROS in treated cells [34,35], while CURC has been reported to prevent such increase [36]. We thus investigated how ROS levels changed in MDA-MB-231 cells exposed to doxorubicin and CURC. As expected, doxorubicin increased intracellular ROS. Free CURC did not prevent such increase, whereas CURC-loaded SLN, either with or without CS, significantly reduced ROS (Figure 3a).

Intracellular ROS can mediate the activation of several transcription factors. Among these redox-sensitive factors, HIF-1α [33,37] and NF-kB [38,39] are well known transcriptional inducers of *mdr1* gene. In MDA-MB-231 cells, HIF-1α activity was increased by doxorubicin, as already reported for other cell types [33], but neither free CURC nor CURC-loaded SLNs affected its activity in doxorubicin-treated and untreated cells (Appendix A). In keeping with the increased ROS levels, doxorubicin-treated cells displayed a higher activation of NF-kB that was not reduced by free CURC and blank SLN. NF-kB activity was instead blunted in doxorubicin-treated cells by CURC-SLN and CURC-CS-SLN (Figure 3b), in keeping with their reduction in ROS levels (Figure 3a).

We hypothesized that the effects of SLNs carrying CURC on the reduced transcription of Pgp was due to a decreased activity of NF-kB. NF-kB is a multimeric transcription factor, whose components (e.g., p50, p65, p52, Rel-A, and c-Rel) form heterodimers characterized by different transcriptional effects [40,41]. Specifically, p65 has been described as binding the *mdr1* promoter and activating the transcription of Pgp [39]. p50/p65 dimer is sequestered in cytoplasm in an inactive form by the inhibitor-kB-α (IkB-α). However, the phosphorylation of IkB-α on serine 32 primes the latter for ubiquitination and proteasomal degradation, freeing p50/p65 dimer to translocate in the nucleus and become an active transcription factor [42]. The master regulator of IkB-α phosphorylation is the IkB-α kinase α/β (IKK-α/β) complex, which, in turn, is activated after being phosphorylated on serine 176 and 180 [43]. Among the multiple kinases activating IKK α/β, there is Akt, upon its phosphorylation on serine 473 [44]. ROS are known activators of Akt and downstream IKK-α/β/NF-kB axis [45] that is often constitutively activated in cancer cells [44,45].

We next investigated which NF-kB components were eventually targeted by CURC-loaded SLNs. Doxorubicin increased p50 (Figure 3c), p65 (Figure 3d), and—to a lesser extent—c-Rel (Appendix A) binding to target DNA sequences. In accordance with previous evidence found in Reference [39], p50, p65, and c-Rel binding activity was reduced by CURC-SLN and CURC-CS-SLN (Figure 3c,d; Appendix A). Of note, CURC-loaded SLNs also reduced the basal activity of p50 and p65 in the absence of doxorubicin, suggesting particularly strong and specific effects of CURC in inhibiting this dimer. To confirm that the nuclear translocation and binding of p50/p65 to DNA was also responsible of the transcription of Pgp, we immunoprecipitated p50 and p65 bound to DNA and amplified the immunoprecipitated DNA with primers specific to the *mdr1* promoter. This ChIP assay indicated that doxorubicin increases the transcription of *mdr1* gene mediated by p50 (Figure 3e) and p65 (Figure 3f). CURC-SLN and CURC-CS-SLN, but not free CURC, decreased p50 and p65-induced transcription of the gene, both in the absence of, or in the presence of, doxorubicin (Figure 3e,f), which confirmed the results of the DNA binding of p50 and p65 (Figure 3c,d).

The expression and activity of upstream activators—Akt, IKKα/β—and inhibitor—IkB-α—of the p50/p65 dimer was measured by immunoblotting. While doxorubicin increased phospho(Ser473)Akt and phospho(Ser176/180)IKKα/β, CURC-loaded SLNs, with or without CS, decreased these phosphorylations, contrarily to free CURC or blank SLN that were devoid of effects. Doxorubicin consistently decreased the total amount of IkB-α. This decrease was restored by CURC-SLN and CURC-CS-SLN, not by free CURC or blank SLN. Notably, both CURC-loaded SLNs reduced phospho(Ser473)Akt, phospho(Ser176/180)IKKα/β, and IkB-α in doxorubicin-treated cells to the same levels detected in untreated cells. Neither free CURC nor SLNs affected the expression of total Akt or IKKα/β. Moreover, no changes were detected in cells not treated with doxorubicin (Figure 3g).

### 3.4. CURC-Loaded SLNs Restore Doxorubicin Sensitivity by Down-Regulating ROS/NF-kB/Pgp Axis in Resistant TNBC Cells Co-Cultured with Macrophages

Tumor associated macrophages (TAMs) are known to play an important role in inducing chemoresistance [46]. To verify whether the chemosensitizing effects of CURC-loaded SLNs were still preserved in the presence of infiltrating macrophages, we set up co-cultures of THP-1-derived macrophages, which phenotypically recapitulated TAM populations [47]. Both CURC-loaded SLNs increased the intracellular accumulation of doxorubicin (Figure 4a) and its cytotoxicity (Figure 4b), and reduced the increase in intracellular ROS elicited by doxorubicin (Figure 4c), as well as the activation of NF-kB (Figure 4d) and the binding of p50 and p65 subunits to the *mdr1* promoter (Figure 4e,f). Consistently, CURC-loaded SLN formulations reduced the levels of Pgp mRNA that were increased by doxorubicin (Figure 4g). Free CURC did not elicit any of these events. CURC-loaded SLNs were less effective in MDA-MB-231 cells co-cultured with macrophages than in MDA-MB-231 cultured alone (Figure 1, Figure 2, Figure 3), but all their effects remained statistically significant (Figure 4a,g).

### 3.5. CURC-Loaded SLNs Are Effective and Safe in Preclinical Models of Pgp-Expressing Mammary JC Tumors

We finally evaluated the efficacy of CURC-SLN and CURC-CS-SLN, alone or in combination with doxorubicin, in mice bearing doxorubicin-resistant/Pgp-expressing JC tumors. While free CURC did not reduce tumor growth (Figure 5a, upper panel) or mass (Figure 5b), both CURC-loaded SLNs decreased the rates of growth (Figure 5a, upper panel). Doxorubicin, alone and in combination with free CURC, was completely ineffective (Figure 5a, lower panel; Figure 5b), in line with the high resistance of these cells detected in vitro (Figure 1e,f). By contrast, the combinations of CURC-loaded SLN, with or without CS, plus doxorubicin, were the most effective in reducing tumor growth (Figure 5a, lower panel) and mass (Figure 5b). These combinations were significantly more effective than doxorubicin, CURC, and CURC-loaded SLNs without doxorubicin (Figure 5a,b).

Furthermore, we measured the hematochemical parameters in the treated animals at the time of sacrifice, to check if there were signs of systemic toxicities. LDH, AST, ALT, and AP were considered indexes of liver toxicity parameters, creatinine as a parameter of kidney, and CPK, CPK-MB, cTnI, and cTnT as parameters of heart toxicity. No treatments affected the liver or kidney-related parameters. As expected, doxorubicin-treated animals had increased CPK, CPK-MB, and cTnT, i.e., typical parameters of heart damage [31]. Neither free CURC nor the CURC-loaded SLNs altered the hematochemical parameters versus the untreated animals. Importantly, they did not worsen the parameters indicative of cardiotoxicity when administered with doxorubicin compared with doxorubicin-treated animals (Table 2).

In all the assays in vitro and in vivo, we did not notice any significant difference between the SLNs with or without CS.

## 4. Discussion

Turmeric or natural curcumoids have been reported to have antioxidant [48], anti-inflammatory [49], and antimicrobial activities [50]. CURC has recently shown anti-tumor properties, relying on the inhibition of oncogenic/pro-survival STAT-3 and NF-kB-dependent pathways, of the pro-invasive factor Sp-1, of tumor-associated inflammation [51,52]. Interestingly, these effects are accompanied by a low toxicity against non-transformed cells [53], suggesting a sort of selectivity against tumor cells.

The main disadvantages of CURC are the poor water-solubility, the unfavorable pharmacokinetic profile, and the easy degradation at slightly alkaline pH, which limit CURC efficacy in clinical practice. To increase the bioavailability of CURC, the use of nanocarriers that may decrease CURC degradation and increase its uptake within tumor cells was proposed [54]. SLN are nanocarriers with high biocompatibility, ideal for the delivery of highly hydrophobic drugs. The recently developed “cold dilution of microemulsion” SLN preparation method [23,24] does not require high temperatures, sonication, use of organic solvents, and pH variations that could negatively influence drug stability or entrapment. Resulting SLNs have small mean diameters ranging from 175 to 190 nm, low polydispersity index (<0.2), a high and reproducible drug entrapment efficacy. Such SLNs were also produced by introducing CS in the microemulsion system, which confers hydrophilic properties to the SLN surface in order to mask or camouflage SLN from the mononuclear phagocytic system (MPS) after parenteral administration. Indeed, it is well known that lipophilic nanoparticles can be quickly opsonized, allowing macrophages of MPS to easily recognize and remove them before they can perform their designed therapeutic function. Moreover, CS is a highly biodegradable and biocompatible material, and has strong adhesive properties to epithelial cells, which should favor the cellular uptake of coated SLNs.

In line with the hypothesis that the entrapment of CURC within SLNs improves its cellular effects, the results of our preliminary comparison of free CURC, CURC-SLN and CURC-CS-SLN indicated that SLN formulations produced the same retention of doxorubicin and doxorubicin-induced toxicity at a concentration (i.e., 5 µM) about 5 to 10-fold lower than free CURC. Moreover, at this concentration, neither blank SLNs nor CURC-SLN and CURC-CS-SLN were cytotoxic for the cells. To achieve the same chemosensitizing efficacy, free CURC had to be used at 25–50 µM, a range of concentrations that elicits cytotoxicity in vitro and is difficult to reach in vivo. These results indirectly suggested that SLNs elicited a higher delivery of CURC within the cells, allowing for the exertion of the chemosensitizing effects at a concentration at which free CURC was ineffective and suggesting the superior efficacy of CURC-loaded SLNs over free CURC.

Mechanistically, CURC delivered by SLN did not act as a competitive inhibitor of Pgp, since it did not increase the retention of rhodamine 123, a classical Pgp substrate, if co-incubated with the dye over a short-term. By contrast, a 24 h pre-incubation of CURC-SLN and CURC-CS-SLN increased the accumulation of both doxorubicin and rhodamine 123. This event was paralleled by the decrease in Pgp mRNA and protein, and was evident in cells treated with doxorubicin. The stronger effects on doxorubicin-treated cells may be explained by the fact that doxorubicin up-regulates Pgp at the transcriptional level [33]. Therein, any inhibitory effect is more evident in cells with an active transcription of the gene. These results look promising because they suggest that CURC-loaded SLNs only reduced Pgp expression in cancer cells after exposure to chemotherapy, without affecting the basal expression of the transporter. This means that the physiological activities of Pgp in non-transformed tissues are likely to be less affected than the activity of Pgp within tumor cells, leading to expect lower, undesired side-toxicities. Moreover, no changes in MRP1 and BCRP mRNA levels were detected, suggesting that the effects of CURC-loaded SLNs were specific for Pgp. Such specificity for Pgp may further contribute to limit the undesired toxicity due to the non-selective inhibition of other ABC transporters.

Unexpectedly, intracellular doxorubicin accumulation and cytotoxicity, levels of Pgp mRNA, and proteins were comparable between CURC-SLN and CURC-CS-SLN. CURC-CS-SLNs are known for their higher hydrophilic properties. It is likely that the good solubility in cell culture medium achieved by CURC-SLN is sufficient to reach the maximal CURC delivery necessary to induce chemosensitizing effects. However, our data suggest that CS coating likely did not change the amount of released CURC within the cells significantly.

To further investigate the mechanisms at the basis of the down-regulation of the doxorubicin-induced Pgp transcription, we focused on three interconnected events elicited by doxorubicin in cancer cells and involved in Pgp up-regulation: The increase in intracellular ROS, and the activation of the transcription factors HIF-1α and NF-kB. Notably, HIF-1α and NF-kB are both Pgp inducers [33,37,38,55] and are activated in response to the increased intracellular ROS [56,57], while CURC is known to prevent oxidative stress [48]. Our data indicated that doxorubicin increased ROS levels, and HIF-1α and NF-kB activity, as expected. CURC-SLN and CURC-CS-SLN reduced ROS and NF-kB activation, without any effects on HIF-1α. We identified p50, p65, and—to a lesser extent—c-Rel, as the NF-kB components inhibited by CURC. Our results are in accordance with previous findings, demonstrating that p65, which commonly dimerizes with p50 [41], is the main NF-kB component binding *mdr1* promoter and activating the transcription of Pgp [40]. By reducing both p65 and p56 binding to *mdr1* promoter, CURC, delivered by SLNs, strongly reduced the levels of Pgp mRNA. Our studies on the signalling upstream NF-kB suggest that the effects of CURC were due to the decreased activation of Akt and IKK-α/β complex upon doxorubicin treatment, which was a consequent of the reduced expression of IkB-α that allowed the nuclear translocation and transcriptional activity of p65/p50 dimer. Since ROS activates the Akt/IKK-α/β/NF-kB axis [44,45], we hypothesized that the *primum movens* of Pgp reduction was the decrease in intracellular ROS induced by CURC and the consequent down-regulation of Akt/IKK-α/β/NF-kB/Pgp pathway. This may explain why the effects of CURC-SLN and CURC-CS-SLN were more pronounced in cells treated with doxorubicin, where there are high levels of intracellular ROS, and are less pronounced in untreated cells, where ROS levels were significantly lower.

The effects of CURC-SLN and CURC-CS-SLN were not cell line or species-specific, since they were shared by both human and murine doxorubicin-resistant cells, suggesting that SLNs achieve their chemosensitizing effects in different mammalian species.

It has already been reported that CURC down-regulates Pgp expression by reducing Akt/NF-kB activation [2]. Indeed, free CURC synergizes with paclitaxel, a typical Pgp substrate in different cancer cell lines [58], which enforces the idea that the chemosensitizing effect of CURC was mediated by a reduction in Pgp transcription. Our approach, however, offers an advancement in this direction, because the use of SLNs allows us to reverse doxorubicin resistance in highly-Pgp expressing cells with low and non-toxic doses of CURC, indicating that the choice of proper biocompatible materials can increase the ratio between anti-tumor benefits/cytotoxicity.

Notably, the chemosensitizing effects of CURC-SLN and CURC-CS-SLN were preserved—although to lesser extent—in doxorubicin-resistant cells, co-cultured with macrophages, which are known for inducing chemoresistance when infiltrating the tumor [46]. The lower efficacy of CURC-loaded SLNs can be explained by the production of several cytokines by TAMs, such as IL-1β and TNF-α that are strong activators of the NF-kB pathway [40]. Such cytokines may act in a paracrine way, synergizing with doxorubicin in activating NF-kB and consequently increasing the amount of Pgp. While less effective, in these tumor-macrophage co-cultures—which represent a stronger model of resistance than cancer cells alone—CURC-loaded SLNs also significantly re-sensitized resistant cells to doxorubicin by the reducing the transcription of Pgp mediated by the ROS/NF-kB axis. These results may suggest that in vivo, where macrophages are an abundant part of the immune-infiltrate [59], the combination of CURC-loaded SLNs and doxorubicin may also be effective against refractory tumors.

Our in vivo experiments support this hypothesis. Indeed, consistent with the anti-tumor properties of CURC [53], SLN formulations loaded with CURC reduced the growth of doxorubicin-resistant JC tumors at dose at which free CURC was ineffective. These results suggest a better bioavailability and delivery to the tumor of CURC when carried by SLNs rather than when administered as a free drug. Most importantly, the effects of CURC-SLN and CURC-CS-SLN were the most pronounced in doxorubicin-treated animals, where the anthracycline alone was ineffective, but its efficacy was rescued by the combination with SLNs. These results represent a proof of concept that CURC, if delivered by SLN, may increase the intratumor accumulation and cytotoxicity of doxorubicin. This is likely a consequence of Pgp down-regulation, an effect elicited on doxorubicin-resistant cells in vitro. Moreover, this experimental set indicated that both SLN formulations did not display signs of systemic toxicities and did not worsen the cardiac damage, indicated by the increase in CPK, CPK-MB, and cTnT induced by doxorubicin. We are aware that, to further ascertain the efficacy and safety of the combination treatments proposed, further investigations are needed. To maximize the anti-tumor efficacy, different regimens (e.g., the co-administration of CURC and doxorubicin, the sequential administration of the drugs, the oscillating administration with the period of drug holidays between each cycle of treatment) should be compared. Similarly, an in depth analysis of the target organs—liver, kidney, heart, lung, bone marrow, and central nervous system—, checking the lack of toxicities beyond hematochemical parameters, is highly desirable to establish the biocompatibility of the SLN. A detailed distribution and pharmacokinetic profile, alongside information on the maximum tolerated doses of CURC-loaded SLNs and doxorubicin, are necessary as well, in order to optimize the scheduled treatments and the route of administration, increasing anti-tumor benefits and safety. These studies are currently ongoing in our group.

As already observed for in vitro assays, there were no differences between CURC-SLN and CURC-CS-SLN, although CS should grant a higher stability and biocompatibility of the nanocarriers. It is likely that CURC-SLN had a good pharmacokinetic profile and suitable characteristics to elicit an anti-tumor property, and a good biocompatibility that avoids side-toxicities. Future in vivo studies might be useful to investigate the role of CS to reduce SLN opsonization and to prolong their plasma half-life.

## 5. Conclusions

Overall, this is the first study validating biocompatible nanocarriers loaded with CURC as new tools able to down-regulate Pgp expression and rescue doxorubicin efficacy against resistant TNBC tumors at lower and non-toxic doses of CURC. These results are particularly relevant because all Pgp inhibitors developed in the recent years repeatedly failed because of their poor specificity and high toxicity, and due to the inhibition of physiological functions of Pgp in non-transformed tissues [60]. SLNs appear to be biocompatible, effective, and safe. Our results are clinically relevant because chemotherapy based on doxorubicin is one of the first therapeutic options in TNBC. Unluckily, this type of breast cancer results in less responsiveness to doxorubicin than other breast cancer types [61], because of the abundant presence of Pgp [62]. Increasing doxorubicin efficacy in TNBC is still an unmet need, but biocompatible SLNs loaded with CURC may help to achieve this goal.

## Figures and Tables

**Figure 1 pharmaceutics-12-00096-f001:**
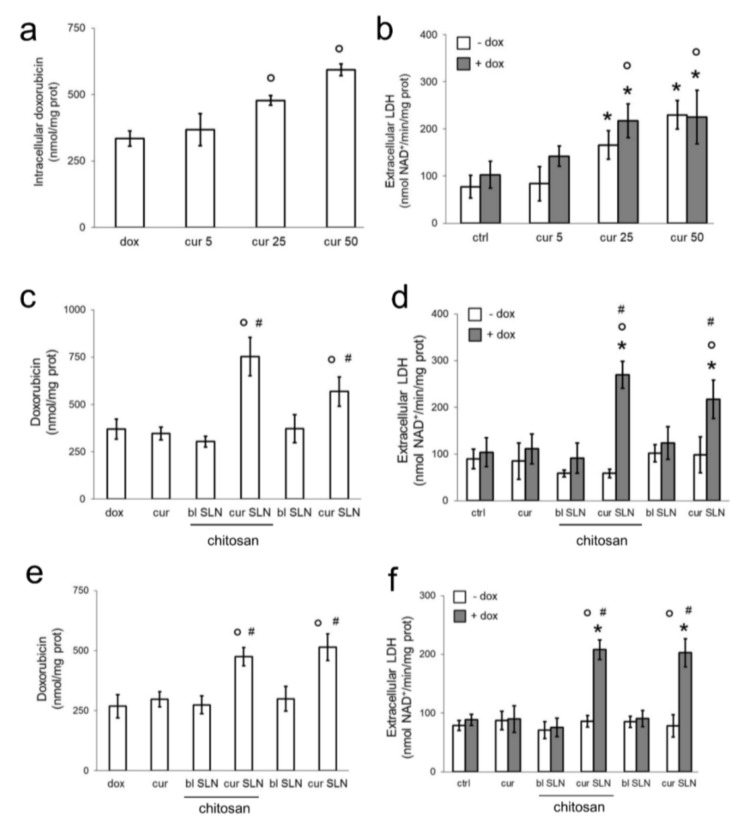
Dose-dependent effects of free CURC and CURC-loaded SLNs on doxorubicin accumulation and cytotoxicity in MDA-MB-231 and JC cells. MDA-MB-231 (panels (**a**–**d**)) and JC (panels (**e**,**f**)) cells were incubated for 24 h with fresh medium (ctrl) or with the 5–25–50 μM CURC (cur), in the absence (−) or presence (+) of 5 μM doxorubicin (dox). (**a**) Intracellular doxorubicin was measured fluorimetrically in duplicates. Data are presented as means ± SD (*n* = 3). ° *p* < 0.01: vs. dox. (**b**) The release of LDH in the extracellular medium was measured spectrophotometrically in duplicates. Data are presented as means ± SD (*n* = 3). * *p* < 0.05: vs. untreated cells (ctrl); ° *p* < 0.05 vs. dox-treated cells. (**c**,**d**) MDA-MB-231 cells were incubated 24 h with fresh medium (ctrl), 5 μM CURC (cur), blank SLN (bl SLN), CURC-loaded SLN (cur SLN, containing 5 μM cur), with or without chitosan. When indicated, 5 μM doxorubicin (dox) was added. (**c**) Intracellular doxorubicin was measured fluorimetrically in duplicates. Data are presented as means ± SD (*n* = 3). ° *p* < 0.001: vs. dox; # *p* < 0.001 vs. cur. (**d**) The release of LDH in the extracellular medium was measured spectrophotometrically in duplicates. Data are presented as means ± SD (*n* = 3). * *p* < 0.001: vs. ctrl; ° *p* < 0.001: vs. dox; # *p* < 0.001 vs. cur. (**e**,**f**) JC cells were incubated as reported in panels C–D. (**e**) Intracellular doxorubicin was measured fluorimetrically in duplicates. Data are presented as means ± SD (*n* = 3). ° *p* < 0.001: vs. dox; # *p* < 0.001 vs. cur. (**f**) The release of LDH in the extracellular medium was measured spectrophotometrically in duplicates. Data are presented as means ± SD (*n* = 3). * *p* < 0.001: vs. ctrl; ° *p* < 0.001: vs. dox; # *p* < 0.001 vs. cur.

**Figure 2 pharmaceutics-12-00096-f002:**
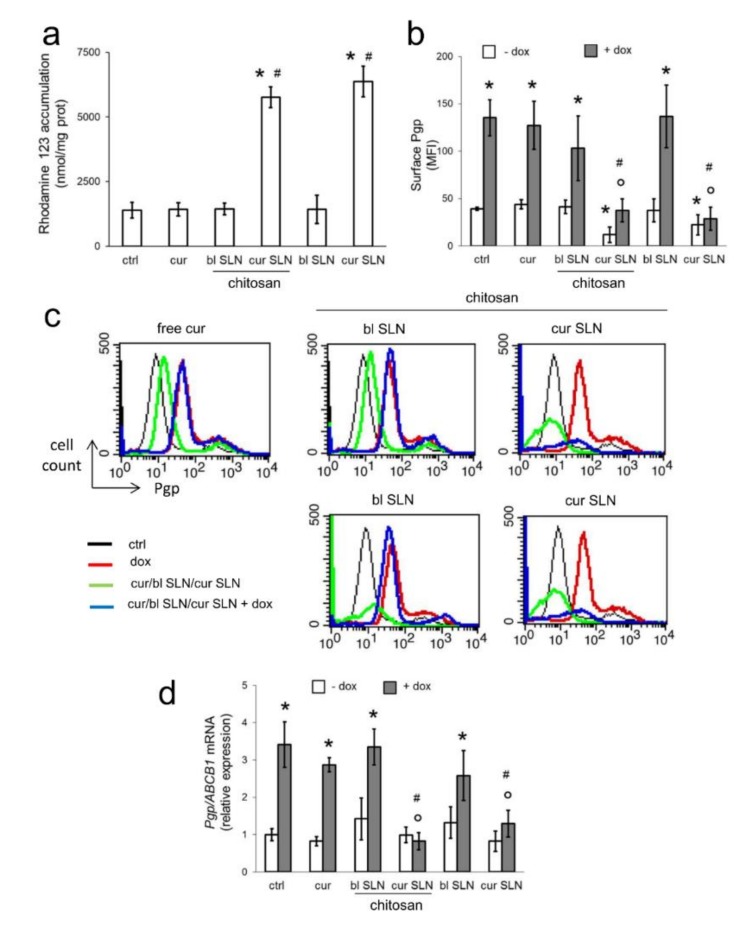
Effects of CURC-loaded SLNs on Pgp activity and expression. MDA-MB-231 cells were incubated for 24 h with fresh medium (ctrl), with 5 μM CURC (cur), blank SLN (bl SLN), CURC-loaded SLN (cur SLN, containing 5 μM cur), and with or without chitosan. When indicated, 5 μM doxorubicin (dox) was added. (**a**) The Pgp substrate rhodamine 123 was added in the last 20 min. The intracellular accumulation of rhodamine 123 was measured fluorimetrically in duplicates. Data are presented as means ± SD (*n* = 3). * *p* < 0.001: vs. ctrl; # *p* < 0.001 vs. cur. (**b**) Pgp on cell surface was measured by flow cytometry; results were expressed as mean fluorescence intensity (MFI). Measurements were performed in triplicate. Data are presented as means ± SD (*n* = 3). * *p* < 0.01: vs. ctrl; ° *p* < 0.001: vs. dox; # *p* < 0.001 vs. cur. (**c**) Representative histograms of one out of three experiments. (**d**) The expression of the Pgp mRNA was measured by qRT-PCR in triplicates. Data are presented as means ± SD (*n* = 3). * *p* < 0.01: vs. ctrl; ° *p* < 0.001: vs. dox; # *p* < 0.001 vs. cur.

**Figure 3 pharmaceutics-12-00096-f003:**
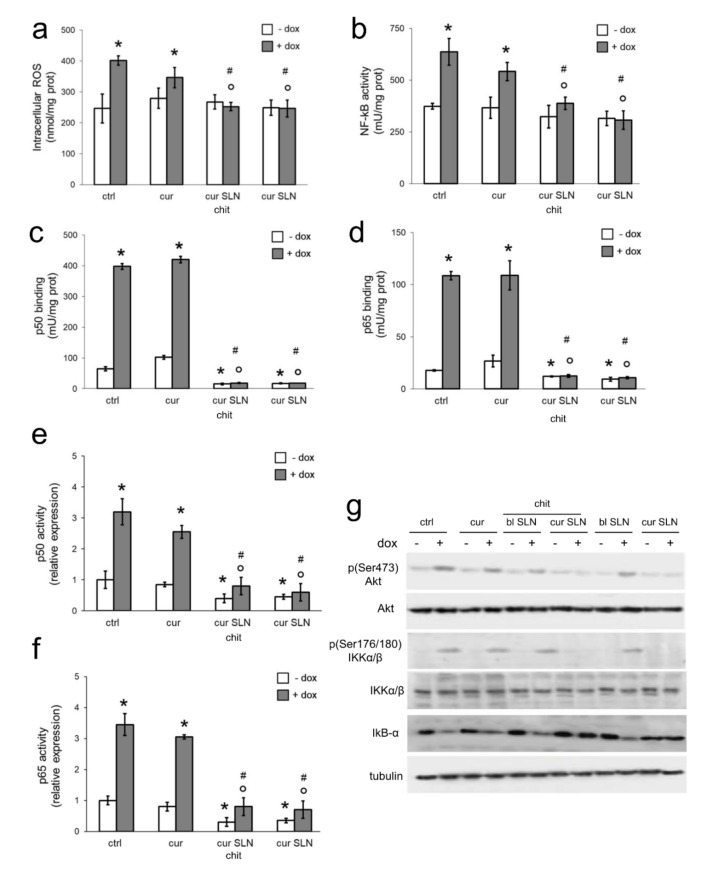
Effects of CURC-loaded SLNs on intracellular ROS, NF-kB activation, and NF-kB-mediated Pgp transcription. MDA-MB-231 cells were incubated with fresh medium (ctrl), 5 μM CURC (cur), and CURC-loaded SLN (cur SLN, containing 5 μM cur), with or without chitosan. When indicated, 5 μM doxorubicin (dox) was added. (**a**) ROS levels were measured fluorimetrically in triplicates. Data are presented as means ± SD (*n* = 3). * *p* < 0.05: vs. ctrl; ° *p* < 0.001: vs. dox; # *p* < 0.01 vs. cur. (**b**) NF-kB activation was measured by ELISA in duplicates. Data are presented as means ± SD (*n* = 3) * *p* < 0.05: vs. ctrl; ° *p* < 0.001: vs. dox; # *p* < 0.01 vs. cur. (**c**,**d**) The binding activity of p50 and p65 was measured by ELISA in duplicates. Data are presented as means ± SD (*n* = 3). * *p* < 0.001: vs. ctrl; ° *p* < 0.02: vs. dox; ^#^
*p* < 0.01 vs. cur. (**e**,**f**) The binding of p50 or p65 to *mdr1* promoter was measured by ChIP. The immunoprecipitated DNA was amplified by qRT-PCR in triplicates and quantified. Data are presented as means ± SD (*n* = 3). * *p* < 0.01: vs. ctrl; ° *p* < 0.001: vs. dox; # *p* < 0.01 vs. cur. (**g**) Cells were lysed and probed with the indicated antibodies. Tubulin was used as the control of the equal protein loading. The figure is representative of one out or three experiments with similar results.

**Figure 4 pharmaceutics-12-00096-f004:**
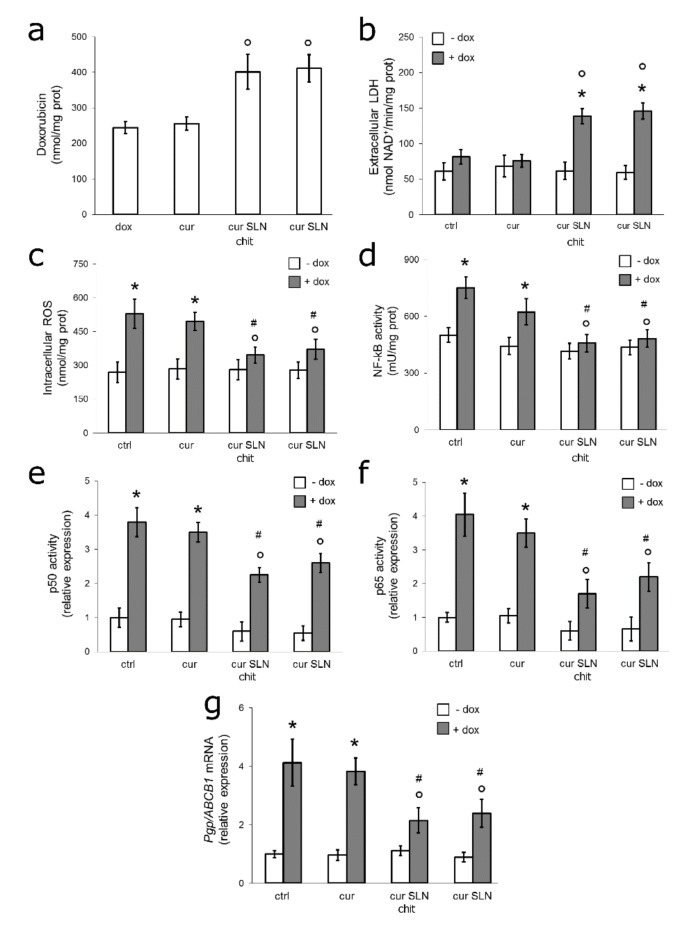
Effects of CURC-loaded SLNs against MDA-MB-231 cells co-cultured with macrophages. Human monocytic THP-1 cells, differentiated into macrophages after 24 h stimulations with 500 nM phorbol-12 myristate 13-acetate, were co-cultured for 24 h with MDA-MB-231 cells (at a 1:10 ratio between macrophages and tumor cells). Co-cultures were maintained in fresh medium (ctrl) or in medium containing 5 μM CURC (cur), CURC-loaded SLN (cur SLN, containing 5 μM cur), with or without chitosan (chit). When indicated, 5 μM doxorubicin (dox) was added. After this incubation time, MDA-MB-231 cells were isolated and subjected to the following analyses. (**a**) Intracellular doxorubicin was measured fluorimetrically in duplicates. Data are presented as means ± SD (*n* = 3). ° *p* < 0.005: vs. dox. (**b**) The release of LDH in the extracellular medium was measured spectrophotometrically in duplicates. Data are presented as means ± SD (*n* = 3). * *p* < 0.001: vs. untreated cells (ctrl); ° *p* < 0.001 vs. dox-treated cells. (**c**) ROS levels were measured fluorimetrically in triplicates. Data are presented as means ± SD (*n* = 3). * *p* < 0.001: vs. ctrl; ° *p* < 0.05: vs. dox; # *p* < 0.02 vs. cur. (**d**) NF-kB activation was measured by ELISA in duplicates. Data are presented as means ± SD (*n* = 3) * *p* < 0.001: vs. ctrl; ° *p* < 0.001: vs. dox; # *p* < 0.02 vs. cur. (**e**,**f**) The binding of p50 or p65 to *mdr1* promoter was measured by ChIP. The immunoprecipitated DNA was amplified by qRT-PCR in triplicates and quantified. Data are presented as means ± SD (*n* = 3). * *p* < 0.001: vs. ctrl; ° *p* < 0.005: vs. dox; # *p* < 0.01 vs. cur. (**g**) The expression of the Pgp mRNA was measured by qRT-PCR in triplicates. Data are presented as means ± SD (*n* = 3). * *p* < 0.001: vs. ctrl; ° *p* < 0.05: vs. dox; # *p* < 0.02 vs. cur.

**Figure 5 pharmaceutics-12-00096-f005:**
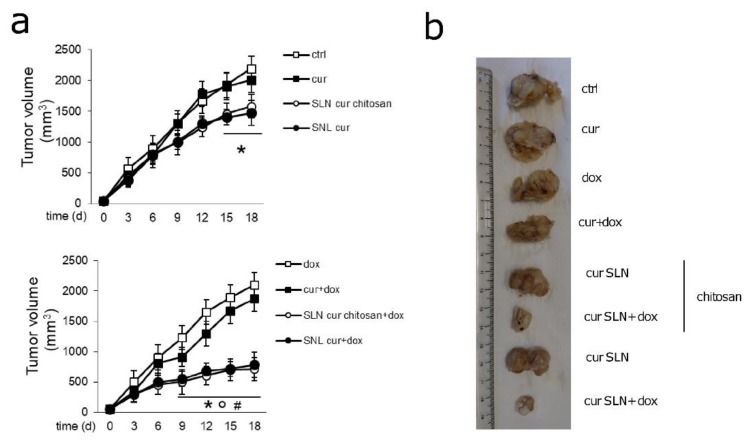
Effects of CURC-loaded SLNs against drug-resistant JC tumors. JC cells were orthotopically implanted into 6 week-old female balb/C mice. When the tumor reached the volume of 50 mm^3^, mice (*n* = 8 mice/group) were randomized and treated, as reported in the following groups, on day 1, 7, and 14 after randomization: (1) Vehicle group (ctrl), treated with 200 µL saline solution intravenously (i.v.); (2) CURC group (cur), treated with 5 mg/kg CURC, dissolved in 200 µL water/10% *v/v* DMSO solution i.v.; (3) doxorubicin group (dox), treated with 5 mg/kg doxorubicin, dissolved in 200 µL water i.v.; (4) CURC + doxorubicin group (cur + dox), treated with 100 µL of water/10% *v/v* DMSO solution containing with 5 mg/kg CURC + 100 µL water solution containing 5 mg/kg doxorubicin; (5) CURC-CS-SLN group, treated i.v. with 200 µL of saline solution of the indicated SLNs containing 5 mg/kg CURC; (6) CURC-SLN group, treated i.v. with 100 µL of saline solution of the indicated SLNs containing 5 mg/kg CURC + 100 µL water solution containing 5 mg/kg doxorubicin; (7) CURC-SLN group, treated i.v. with 100 µL of saline solution of the indicated SLNs containing 5 mg/kg CURC; and (8) uncoated/CURC-loaded SLN + doxorubicin group (cur SLN + dox), treated i.v with 200 µL of saline solution of the indicated SLNs containing 5 mg/kg CURC + 100 µL water solution containing 5 mg/kg doxorubicin. (**a**) Tumor growth was monitored daily by caliper measurements. Data are presented as means ± SD. * *p* < 0.01: vs. ctrl; * *p* < 0.001: vs. dox; # *p* < 0.001: vs. cur. (**b**) Photographs of representative tumors of each group.

**Table 1 pharmaceutics-12-00096-t001:** Compositions of the CURC-loaded SLNs synthesized.

Components	CURC-SLN (*w/w* %)	CURC-CS-SLN (*w/w* %)
TL	4.22	-
Cholesterol	-	3.22
CS	-	0.43
Epikuron^®^200	10.57	16.12
s-EA	14.09	-
s-BL	-	9.54
CURC	0.63	0.64
Cremophor^®^RH60	3.52	9.54
NaTC	3.53	9.03
1,2 Propanediol	14.09	-
BenzOH	-	6.71
s-W	49.35	44.77

TL: trilaurin; s-EA: pre-saturated ethyl acetate; s-BL: pre-saturated butyl lactate; CURC: curcumin; CS: chitosan; NaTC: sodium taurocholate; BenzOH: benzyl alcohol; s-W: presaturated water.

**Table 2 pharmaceutics-12-00096-t002:** Hematochemical parameters of the treated mice ^1^.

**−Doxorubicin**	**Ctrl**	**CURC**	**CURC-CS-SLN**	**CURC-SLN**
LDH (U/L)	7091 ± 639	7189 ± 409	6781 ± 1021	7112 ± 678
AST (U/L)	89 ± 54	101 ± 43	121 ± 48	99 ± 29
ALT (U/L)	38 ± 8	41 ± 8	37 ± 10	34 ± 8
AP (U/L)	134 ± 49	139 ± 41	129 ± 29	139 ± 18
Creatinine (mg/L)	0.056 ± 0.005	0.062 ± 0.010	0.062 ± 0.009	0.054 ± 0.008
CPK (U/L)	314 ± 99	312 ± 83	382 ± 56	334 ± 39
CPK-MB (ng/mL)	0.109 ± 0.058	0.118 ± 0.032	0.129 ± 0.045	0.118 ± 0.041
cTnI (pg/mL)	1.011 ± 0.082	1.019 ± 0.032	1.008 ± 0.062	1.024 ± 0.029
cTnT (pg/mL)	2.182 ± 0.213	2.178 ± 0.101	2.189 ± 0.123	1.897 ± 0.162
**+doxorubicin**	**Ctrl**	**CURC**	**CURC-CS-SLN**	**CURC-SLN**
LDH (U/L)	7561 ± 761	7192 ± 506	7821 ± 821	6523 ± 801
AST (U/L)	103 ± 44	132 ± 45	105 ± 36	137 ± 89
ALT (U/L)	36 ± 15	41 ± 18	39 ± 13	56 ± 19
AP (U/L)	138 ± 25	167 ± 56	139 ± 44	168 ± 41
Creatinine (mg/L)	0.083 ± 0.009	0.082 ± 0.009	0.093 ± 0.011	0.093 ± 0.010
CPK (U/L)	556 ± 89 *	571 ± 89 *	562 ± 81 *	504 ± 81 *
CPK-MB (ng/mL)	0.302 ± 0.71 *	0.287 ± 0.045 *	0.297 ± 0.062 *	0.322 ± 0.016 *
cTnI (pg/mL)	1.021 ± 0.039	1.033 ± 0.046	1.031 ± 0.067	1.019 ± 0.052
cTnT (pg/mL)	3.197 ± 0.209 *	2.882 ± 0.172 *	2.904 ± 0.209 *	2.821 ± 0.178 *

^1^ Balb/C mice (*n* = 8 mice/group) were treated as described in Figure 4. Blood was collected immediately after euthanasia and analyzed for lactate dehydrogenase (LDH), aspartate aminotransferase (AST), alanine aminotransferase (ALT), alkaline phosphatase (AP), creatinine, creatine phosphokinase (CPK) and CPK-MB, cardiac troponin I (cTnI) and T (cTnT). Data are presented as means ± SD. * *p* < 0.05: vs. ctrl group.

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
