# Peer review of "Curcumin-Loaded Solid Lipid Nanoparticles Bypass P-Glycoprotein Mediated Doxorubicin Resistance in Triple Negative Breast Cancer Cells"

_pharmaceutics, 2020, doi:10.3390/pharmaceutics12020096_

Round 1

Reviewer 1 Report

This is  very interesting and important contribution for the chemotherapy of multidrug resistant tumour cells. It is well written with the exception of some minor errors. I only think that the experimental methods could be described with more detail, namely, an explanation of how Curcumin is incorporated into de SLNs is missing. Also some references are missing. All this aspects are indicated in an annotated pdf file that I attach (the annotations were made using Okular software in KDE/linux).

Author Response

According to the annotations of the Reviewer on the pdf file, we:

- provided the definition of neo-adjuvant chemotherapy (line 53)

- corrected the sentence at line 68

- corrected the size of SLN (line 68): indeed the mean diameter of SLN ranges from 800 nm to100 nm, with a mean diameter of 400 nm. The SLNs used in the present work had a mean size of 200 nm

- included the detailed procedure of the preparation of Curcumin (CURC)-loaded SLN in the Materials and Methods section (line 95). We also added a new table (Table 1 in the revised version)

- detailed how the calibration curve for doxorubicin measurement was performed, specifying that the drug had the same fluorescence spectrum in the acellular solution used for the calibration curve and in the cell lysates (line 149)

- added a reference for the DCFDA-AM method (line 205)

- added a reference for the euthanasia method (line 256)

- corrected the sentence at line 271

- included the dose-dependence experiments of cytotoxicity exerted by blank SLNs, i.e. SLNs without CURC, as new Supplementary Figure S2 (line 281).

Reviewer 2 Report

The authors investigated whether curcumin loaded in biocompatible solid lipid nanoparticles could sensitize triple negative breast cancer cells to doxorubicin treatment via inhibition of the activitity and the expression P glycoprotein transporter. Nevertheless, data are only preliminary as all mechanistic studies were performed only in vitro and only in the presence of breast cancer cell lines. To increase the value of these results other in vitro models must be used such as co-cultures of cancer cells and immune cells as immune cells like macrophages are known that can modulate the sensitivity of the tumor cells to cytotoxic drug treatments.

The in vivo data are only preliminary and not sufficient to support the main hypothesis of this research. Moreover, the authors tried only a type of regimen of administration of doxorubicin together with solid lipid nanoparticles with curcumin. They must try a sequential treatment based on a pretreatment with olid lipid nanoparticles with curcumin followed by doxorubicin administration.

Author Response

Comments and Suggestions for Authors

1) The authors investigated whether curcumin loaded in biocompatible solid lipid nanoparticles could sensitize triple negative breast cancer cells to doxorubicin treatment via inhibition of the activitity and the expression P glycoprotein transporter. Nevertheless, data are only preliminary as all mechanistic studies were performed only in vitro and only in the presence of breast cancer cell lines. To increase the value of these results other in vitro models must be used such as co-cultures of cancer cells and immune cells as immune cells like macrophages are known that can modulate the sensitivity of the tumor cells to cytotoxic drug treatments.

We thank the Reviewer for the valuable suggestion.

We set up a co-culture system using the human monocytic THP-1 cells that can be easily differentiated and propagated as macrophages, reproducing a “tumor-associated macrophages” (TAMs) phenotype [Genin et al, BMC Cancer 2015, 15, 577]. THP-1 cells were seeded in RPMI-1640 complete medium at a density of 1 x 10 5 cells/well in a 6-well-plate. Cells were stimulated with 500 nM phorbol-12 myristate 13-acetate (PMA) for 24 h, in order to favor their differentiation into macrophages [Yuan et al, Oncology Lett.. 2019, 18: 1840-1846]. Floating cells were removed by washing the dish with PBS twice, attached cells – i.e. macrophage-differentiated cells – were rinsed with fresh medium. 1 x 10 6 MDA-MB-231 cells (i.e. a 1:10 ratio between macrophages and tumor cells) [Long et al, Bio Protoc. 2018, 8(8): doi:10.21769/BioProtoc.2815.] were added to the monolayer of macrophages. Co-cultures were incubated for 24 h with fresh medium (ctrl), 5 μM curcumin (CURC), CURC-loaded SLN (containing 5 μM CURC), with or without chitosan, in the presence or absence of 5 μM doxorubicin. Cells were detached, macrophages were removed using the Pan Monocyte Isolation Kit (Miltenyi, Bergisch Gladbach, Germany), as per manufacturer’s instructions. The intracellular doxorubicin accumulation, the cytotoxicity - evaluated as release of lactate dehydrogenase (LDH) -, the intracellular ROS levels, the activity of p50/p65 NF-kB dimer, measured by ELISA and ChIP on mdr1 promoter, the levels of Pgp mRNA were evaluated on purified MDA-MB-231 cells derived from the co-cultures.  

As shown in the new Figure 4, both CURC-loaded SLN formulations increased the intracellular accumulation of doxorubicin and its cytotoxicity, reduced the increase in intracellular ROS elicited by doxorubicin, as well as the activation of NF-kB and the binding of p50/p65 subunits to mdr1 promoter. Consistently, CURC-loaded SLNs reduced the levels of Pgp mRNA that was instead increased by doxorubicin. CURC did not elicit any of these events. In cells derived from co-cultures, CURC-loaded SLNs were less effective than in MDA-MB-231 cells cultured alone (Figures 1-2-3). This trend could be explained by the production of several cytokines by macrophages, such as IL-1β and TNF-α, that are strong activators of NF-kB pathway [Orlowski and Baldwin, Trends Mol Med 2002, 8, 385–389]. Such cytokines may act in a paracrine way, synergizing with doxorubicin in activating NF-kB and consequently increasing the amounts of Pgp. However, also in tumor-macrophage co-cultures – that represent a stronger model of resistance than cancer cell alone – it is noteworthy that CURC-loaded SLNs significantly sensitized resistant cells to doxorubicin by reducing the ROS/NF-kB axis and the transcription of Pgp. These results may suggest that also in vivo, where macrophages are known to play an important role in inducing tumor progression and chemoresistance [Yin et al, Clin Cancer Res. 2017, 23, 7375–7378; Komohara and Takeya, J Pathol. 2017, 241, 313-315], the combination of CURC-loaded SLNs and doxorubicin may be effective against refractory tumors.

We modified Materials and Methods (line 134), Results (line 410) and Discussion (line 579) accordingly. We add one new Figure and five new references.

2) The in vivo data are only preliminary and not sufficient to support the main hypothesis of this research. Moreover, the authors tried only a type of regimen of administration of doxorubicin together with solid lipid nanoparticles with curcumin. They must try a sequential treatment based on a pretreatment with olid lipid nanoparticles with curcumin followed by doxorubicin administration.

We agree that our in vivo data are preliminary, but they represent the proof of concept of antitumor efficacy and safety of the proposed combination therapy, based on the association of CURC-loaded-SLN and doxorubicin. Different treatment regimens (e.g. sequential treatments, oscillating treatments including drug-holiday periods), alongside with studies on the pharmacokinetic profile, the maximum-tolerated doses and the cardio-safety of the combination treatment will be the subject of a dedicated work. We specified these points and the future perspectives in the Discussion (line 596).  

Round 2

Reviewer 2 Report

I agree the manuscript to be accepted in the present form.